# Anxiety and associated factors among medical and surgical patients in Ethiopia: A systematic review and meta-analysis

**Gidey Rtbey**◉*, **Milen Mihertabe, Fantahun Andualem, Mamaru Melkam, Girmaw Medfu Takelle, Techilo Tinsae, Setegn Fentahun**

Department of Psychiatry, College of Medicine and Health Sciences, University of Gondar, Gondar, Ethiopia

* gidur2006@gmail.com

**Data Availability Statement:** All the data is available in the paper.

**Funding:** The author(s) received no specific funding for this work.

## Abstract

### Background

Individuals diagnosed with chronic medical conditions and patients appointed to undergo surgery face various degrees of anxiety as a result of doubts related to the outcome of surgery, and the psycho-socioeconomic costs of the medical illness. This can affect the treatment process and even the outcome of patients with medical and surgical cases. Though different studies were conducted on anxiety and associated factors among medical and surgical patients in Ethiopia, the findings were found to be inconsistent and had a wide discrepancy. So, this systematic review and meta-analysis estimated the pooled effect size of anxiety among this population and guides to plan appropriate intervention at a national level.

### Methods

Studies conducted on anxiety and associated factors among medical and surgical patients in Ethiopia were included. Data was extracted using Microsoft Excel and analyzed using STATA version 11. The random-effects model was used to estimate the pooled effect size of anxiety and its determinants with 95% confidence intervals. Funnel plots and Egger's regression tests were employed to check publication bias. Sub-group and sensitivity analyses were also conducted.

### Results

The pooled prevalence of anxiety among medical and surgical patients in Ethiopia was found to be 48.82% with a 95% CI (42.66, 54.99). Being female[OR = 2.84(2.02, 4.01)], fear of death [OR = 2.93(1.57, 5.50)], and history of surgery[OR = 0.42(0.27, 0.065)], among surgical patients and being female[OR = 2.35(1.94, 2.850], having poor social support[OR = 2.22(1.62, 3.05)], perceived stigma[OR = 4.25(1.97, 9.18)] and family history of mental illness[OR = 1.86(1.21, 2.86)] among medical patients were significantly associated with anxiety in this systematic review and meta-analysis.

**Competing interests:** The authors have declared that no competing interests exist.

**Abbreviations:** AIDS, Acquired Immuno-deficiency Syndrome; AOR, Adjusted Odd Ratio; BAIS, Beck Anxiety Inventory Scale; CI, Confidence Interval; DASS-21, Depression Anxiety and Stress Scale; GAD-7, Generalized Anxiety Disorder 7-item; HADS, Hospital Anxiety and Depression Scale; HAS, Hospital Anxiety Scale; PITI, Preoperative Intrusive Thought Inventory; PRISMA, Preferred Reporting Items for Systematic Review and Meta-Analysis; SNNPs, South Nation Nationality and Peoples; S-STAI, State-Trait Anxiety Inventory Scale; USA, United States of America; WHO, World Health Organization.

## Conclusion and recommendation

The pooled prevalence of anxiety among medical and surgical patients in Ethiopia was found to be high. Therefore, it would be good for professionals to screen patients for anxiety besides managing their medical or surgical cases to detect them early and address them.

## Introduction

Anxiety is described as a feeling of emotional unease, distress, apprehension, or apprehensive concern that is subjective, accompanied by somatic and autonomic symptoms, and that could impair an individual's functioning [1]. In addition to being a natural emotional response to any life stress, anxiety can also have physical and psychological components [1,2].

Surgery is a potentially life-threatening procedure that makes the patient to be directly physically restrained. Patients who are scheduled to undergo surgery may have anxieties related to fear of dying, discomfort after the procedure, and apprehension that they won't be able to wake up from anesthesia [3]. Regardless of the type of surgery to be performed, it is a procedure that could induce worry, which could lead patients to suffer varying degrees of anxiety [2,4].

Aside from surgical procedures, one of the most common and debilitating neuropsychiatric symptoms associated with chronic medical illnesses such as cancer is the development of anxiety symptoms [5]. In the beginning, cancer patients feel shock or denial, followed by emotional turmoil, anxiety, lack of concentration, difficulty falling asleep, loss of appetite, irritability, and intrusive worries about the future [6]. Those living with HIV/AIDS and other chronic illnesses may experience anxiety as they attempt to manage the consequences of the diagnosis and deal with the difficulties, such as shorter life expectancies, difficult treatment regimens, stigma, and loss of social and familial support [7].

According to a meta-analysis, preoperative anxiety was present in 48% of surgical patients globally [8], with low and middle-income countries(LMICs) accounting for 55.7% of these cases [9]. In Ethiopia, the prevalence of anxiety among surgical patients ranges between 47% and 70% [10,11]. On the other hand, manifestations of anxiety symptoms among patients with chronic medical illnesses like cancer, HIV/AIDS, hypertension, diabetes mellitus, and kidney disease range between 28.5% and 64.9% [12,13].

The consequences of anxiety symptoms among surgical patients are acute myocardial infarction, heart failure, pulmonary edema, high readmission rate, poor quality of life, and high rate of cardiac mortality which correlate with high postoperative pain, increased analgesic and anesthetic consumption, prolonged hospital admission, adverse influence during anesthetic induction and patient recovery and decrease patient satisfaction with perioperative care [14–17].

In individuals with chronic medical conditions, anxiety symptoms can have a variety of detrimental effects on their health compared to the general population [18]. Untreated and unrecognized anxieties have negative repercussions such as medication non-adherence, rapid disease progression, and poor health outcomes [19]. Because anxiety can be so crippling, it can affect a person's daily functioning and quality of life, especially in the case of diabetes mellitus which increases the mortality rate [20]. Besides this, the comorbidity of anxiety symptoms has an adverse economic impact on the patient, family, and the community at large due to the costs involved with diagnosis, treatment, and lost productivity [21].

Different studies have shown that the type of surgery, the patient's gender, their interaction with the medical personnel, having had surgery before, and sensitivity to stressful situations

are all factors that contribute to anxiety in surgical patients [22–24]. Furthermore, concern about the results of surgery (29.3%), worry about the recovery process (19.5%), and complications of the surgery (11.4%) were the most often reported significant causes of preoperative anxiety [25]. While being widowed or divorced, being a female sex, low educational level, unemployment, high body mass index, low income, poor social support, substance abuse, and non-adherence to medication were all linked to anxiety in patients with chronic medical conditions [20,26–28].

Medical and surgical patients' health status is expected to be significantly improved as a result of the resources allocated to their health care delivery within the country. Unluckily, some circumstances compromise patients' outcomes and prevent us from accomplishing these ultimate goals. Anxiety symptoms are one of the obstacles that lead to poor outcomes among this segment of the population. To address such kind of hindrances, evidence about the burden of anxiety and its determinants is a crucial input at a national level. This study was aimed at determining the pooled effect size of anxiety and factors associated with it among chronically ill medical patients and surgical patients in Ethiopia. So, having such evidence-based data will be helpful to plan and carry out possible interventions in the future.

## Methods

### Protocol and registration

The protocol for this systematic review and meta-analysis was registered in the International Prospective Register of Systemic Review (PROSPERO) (ID = CRD42023427023).

### Study design

Systematic review AND Meta-analysis were performed according to the Preferred Reporting Items for Systematic Review and Meta-Analysis (PRISMA-P 2020) standard [29] (S1 File).

### Searching strategies

All available research articles were reviewed using the following databases: PubMed/MEDLINE, Scopus, African-Wider, PsycINFO, EMBASE, Google Scholar, AJOL, and World Health Organization (WHO) reports. In addition, a search of other gray literature was tried by searching Google, Google Scholar, and other internet search engines to search for additional articles until the end of May 2023. The search terms used were anxiety and associated factors among patients with chronic medical conditions and those undergoing surgery OR other Medical Subject Heading (Mesh), keywords, and free text search terms. The authors used several phrases, such as anxiety OR anxiety condition, medical and surgical patients, and combined terms by utilizing Boolean operators while searching for the already available literature. The terms that were used to search the articles were ("anxiety OR anxiety disorder") AND ("associated factors" OR "determinants" AND "chronic medically ill patients" AND "surgical patients" OR "patients undergoing surgery" AND "Ethiopia").

### Eligibility criteria

All peer-reviewed journal articles and grey literature that addressed anxiety and associated factors among patients with chronic medical illness and patients undergoing surgery were included in this systematic review and meta-analysis. All observational studies using various study designs on anxiety among medical and surgical patients in Ethiopia were included. Studies that lacked an abstract, didn't have a full text or were difficult to get the necessary data were excluded from this systematic review and meta-analysis.

## Outcome variable measurement tool

The outcome variable of this systematic review and meta-analysis was anxiety, which was assessed using different tools in the included studies. Among the included studies 11 of them were assessed using the hospital anxiety depression scale (HADS), 7 were using the state-trait anxiety inventory scale (S-STAI), and the rest 5 used other tools (PITI, HAS, GAD-7, DAS, and BAIS). The 14-item HADS questionnaire is used to check for symptoms of anxiety and depression. Its internal consistency for anxiety was 0.78 and it was approved for local usage in Ethiopia. The scales use a cut-off point of $\geq 8$ for anxiety [30]. State-Trait Anxiety Inventory Scale (S-STAI) was validated in Ethiopia [31]. The State-Trait Anxiety Inventory Scale (S-STAI) features two subscales with a total of 40 items, 20 of which are assigned to the S-Anxiety and T-Anxiety subscales. The State Anxiety Scale (S-Anxiety), which measures subjective feelings of dread, tension, uneasiness, worry, and activation/arousal of the autonomic nervous system, assesses the current state of anxiety by asking respondents how they feel "right now." The Trait Anxiety Scale (T-Anxiety) measures relatively constant features of "anxiety proneness," such as overall feelings of calm, assurance, and security [32].

## Data extraction

After carefully examining the papers' titles, abstracts, and full texts, the pertinent data was extracted using a Microsoft Excel Spreadsheet. All researches that have been endorsed by the reviewers were included. The data was extracted using the authors' names, years of publications, study design, study region, case type (medical or surgical), and sample size from the included articles. Additionally, the estimated combined effects of anxiety and its associated factors were retrieved along with their 95% confidence intervals. GR conducted the primary data extraction, and then FA assessed the extracted data independently. Disagreements and discrepancies were resolved through discussion with the third author MM (the fourth author) (S3 File).

## Quality appraisal of the selected studies

To independently assess the quality of the research articles included in this systematic review and meta-analysis, two authors (SF and MM (the second author)) used the standard instrument. The Joanna Briggs Institute (JBI) assessment tool was used to assess the quality of the studies [33]. This quality measuring instrument was developed for articles that reported prevalence data and observational studies. The quality assessment of the articles was the same because of the same study design for all (cross-sectional). Articles with a final rating scale of five and above out of nine points were included in this systematic review and meta-analysis. Differences between reviewers in quality ratings were resolved in a discussion with another third author (GR) to reach a common agreement (S2 File).

## Data processing and analysis

The extracted data was exported to STATA version-11 statistical software for analysis. A random-effects model was used to obtain the pooled effect size and the effect size of all articles with 95% confidence intervals. A forest plot was employed in the graphical summary to estimate the pooled effect size and weight of each recruited study with a 95% confidence interval. The degree of heterogeneity among the included articles was assessed using the index of heterogeneity ($I^2$ statistics) [34]. Funnel plot analysis and Egger's weighted regression tests were also performed to check publication bias [35,36]. After confirming the presence of publication bias in Egger's test (0.039) trim and fill analysis was done to adjust the observed publication

bias [37]. Sub-group analysis employing the study region, case type (medical or surgical), and assessment tool used was conducted.

## Results

### Study selection process

Using various search databases, 1990 studies were first identified, and the extra records led to the addition of 90 more studies. Due to duplication, 745 papers were removed, and 1190 studies were excluded because their titles were irrelevant or their goals were unrelated to this investigation. Out of the 145 studies that underwent full-text evaluation, only 23 matched the qualifying requirements for this systematic review and meta-analysis (Fig 1).

### Characteristics of studies included

This systematic review and meta-analysis included a total of 23 studies conducted on anxiety and associated factors among chronic medical and surgical patients in Ethiopia. Out of these nine articles were conducted in Amhara, six in Addis Ababa, four in Oromia, the final three in SNNPs, and one in the Somalia region (Table 1).

### Meta-analysis

The pooled prevalence of anxiety among chronic medically ill and surgical patients in Ethiopia was found to be 48.82% with a 95% CI (42.66, 54.99) (Fig 2). In this systematic review and meta-analysis, the pooled prevalence of anxiety in Amhara, Addis Ababa, Oromia, SNNPs, and Somalia region was 48.7%, 49.7%, 54.11%, 47.14%, and 29.8% respectively (Table 3).

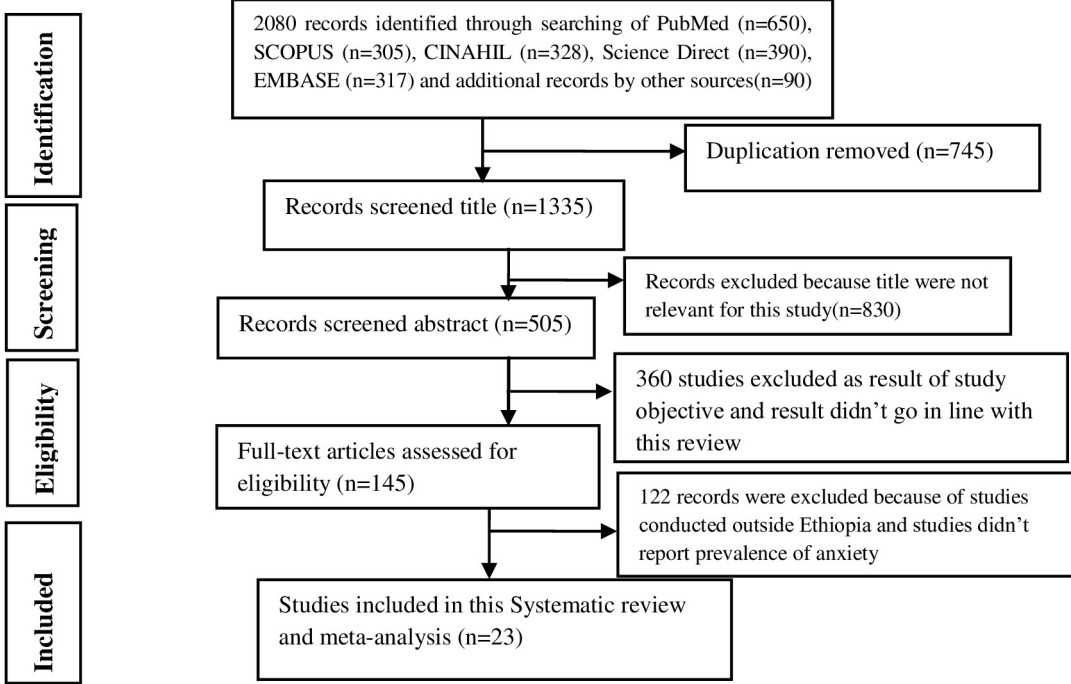

**Fig 1. Flow chart describing the selection of studies conducted on anxiety among medical and surgical patients in Ethiopia.**

**Table 1. Characteristics of studies selected in this systematic review and meta-analysis on anxiety and associated factors among medical and surgical patients in Ethiopia.**

| Authors' name | Region | Sample size | Prevalence of anxiety (%) |
|---|---|---|---|
| Afrassa et al., 2022 [11]. | Addis Ababa | 267 | 70.4 |
| Bedaso and Ayalew, 2019 [10]. | SNNPs | 407 | 47 |
| Ferede et al., 2022 [38]. | Amhara | 374 | 63 |
| Mulugeta et al., 2018 [39]. | Amhara | 353 | 61 |
| Nigussie et al., 2014 [40]. | Oromia | 239 | 70.3 |
| Shewangzaw Engda et al., 2022 [41]. | Amhara | 354 | 53.6 |
| Srahbzu et al., 2018 [42]. | Addis Ababa | 407 | 39.8 |
| Woldegerima et al., 2018 [43]. | Amhara | 181 | 59.6 |
| Wondmieneh, 2020 [44]. | Amhara | 216 | 48.3 |
| Aberha et al., 2016 [12]. | Addis Ababa | 417 | 28.5 |
| Atinafu et al., 2022 [13]. | Addis Ababa | 171 | 64.9 |
| Ayalew et al., 2022 [45]. | SNNPs | 415 | 60 |
| Belete et al., 2014 [46]. | Amhara | 443 | 22.2 |
| Duko et al., 2019 [47]. | SNNPs | 363 | 34.4 |
| Edmealem and Olis, 2020 [48]. | Amhara | 404 | 32 |
| Endeshaw et al., 2022 [49]. | Amhara | 423 | 57.1 |
| Hajure et al., 2020 [50]. | Oromia | 423 | 61.8 |
| Nigussie et al., 2023 [51]. | Oromia | 421 | 40.4 |
| Tesfaw et al., 2016 [52]. | Addis Ababa | 417 | 32.4 |
| Tesfaw et al., 2022 [53]. | Addis Ababa | 590 | 62.7 |
| Tibebu et al., 2023 [54]. | Amhara | 423 | 42.1 |
| Tiki, 2017 [55]. | Oromia | 423 | 44.2 |
| Yousuf et al., 2020 [56]. | Somalia | 357 | 28.9 |

## Heterogeneity and publication bias

Heterogeneity was identified in this systematic review and meta-analysis with $I^2$ of 97.3% and a P-value of 0.001. To determine whether the chosen articles might have a publication bias, a funnel plot was used. The absence of publication bias in the included papers is indicated by the funnel plot's placement seemingly symmetrical in distribution (Fig 3). However, egger's regression test confirmed the presence of publication bias (P = 0.039) (Table 2). To adjust this, publication bias trim and fill analysis was employed (Fig 4).

## Subgroup analysis

After confirming the presence of heterogeneity, a subgroup analysis based on study region, case type (medical and surgical), and assessment tools was conducted. In Amhara region, the pooled prevalence of anxiety among medical and surgical patients was 48.7%, 49.7% in Addis Ababa, 54.11% in Oromia, and 42.59% in other regions (SNNPs and Somalia). Analysis based on the case type revealed that pooled prevalence was 56.97% among surgical and 43.63% among medical patients whereas 57.55%, 44.66%, and 45.85% were pooled prevalence in studies that used S-STAI, HADS, and other assessment tools respectively (Table 3).

## Sensitivity analysis

By omitting one study at a time, a sensitivity analysis was conducted to see how each study's findings affected the overall prevalence of anxiety in this systematic review and meta-analysis.

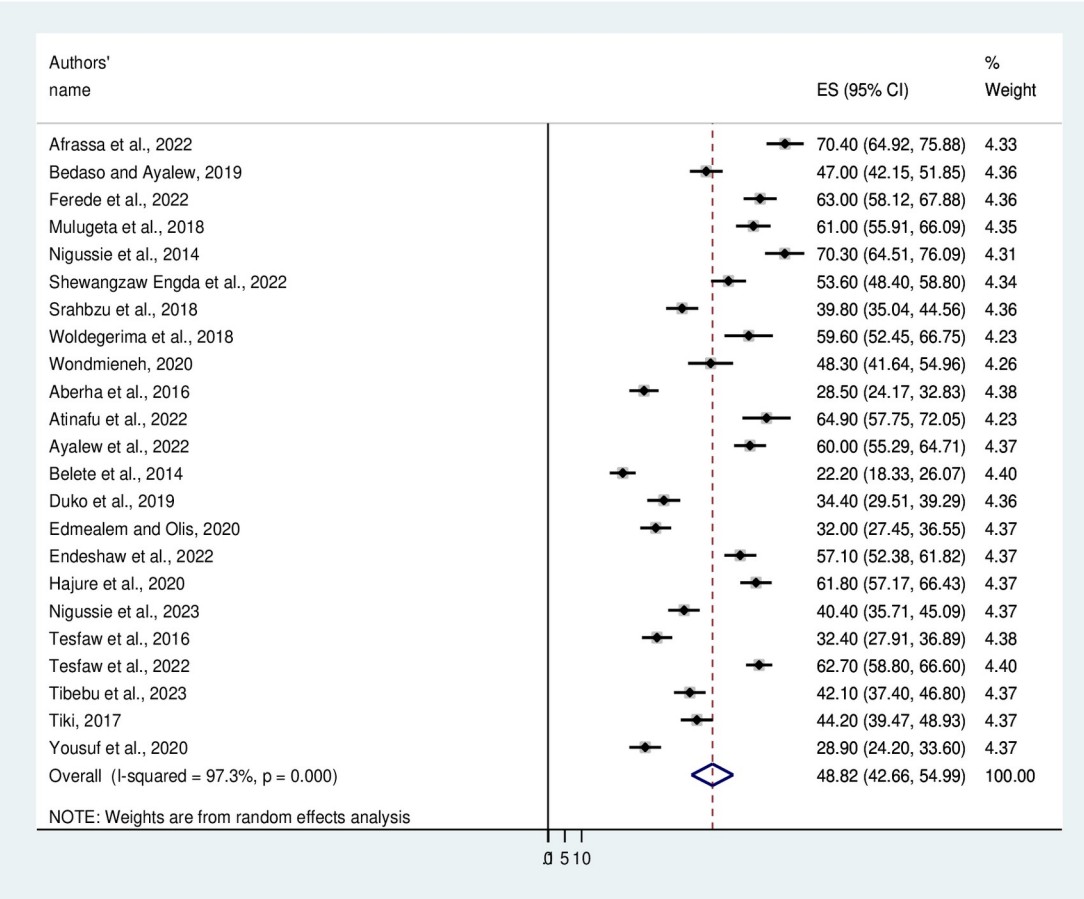

**Fig 2. A Forest plot that shows the pooled level of anxiety among medical and surgical patients in Ethiopia.**

The sensitivity analysis showed that all values were almost within the estimated 95% CI, indicating that the prevalence of this meta-analysis was not significantly affected by the removal of a single study (Fig 5).

## Determinants of anxiety among medical and surgical patients

In this systematic review and meta-analysis, factors contributing to anxiety among surgical patients in Ethiopia were identified. Being female sex was found to be 2.84 times more likely to have anxiety than males [OR = 2.84(2.02, 4.01)]. Having a fear of death among surgical patients was also a contributing factor to anxiety in this study. Participants who had a fear of death were 2.93 times higher in developing anxiety compared to their counterparts [OR = 2.93 (1.57, 5.50)].

The other variable associated with anxiety was having a history of surgery. Surgical patients who had a history of surgery were 58% less likely to have preoperative anxiety than participants who didn't have a history of surgery [OR = 0.42(0.27, 0.065)] (Fig 6).

Regarding variables associated with anxiety among chronic medical patients in this systematic review and meta-analysis, being female was higher in developing anxiety than males [OR = 2.35(1.94, 2.85)]. The other predictive variable that was significantly associated with

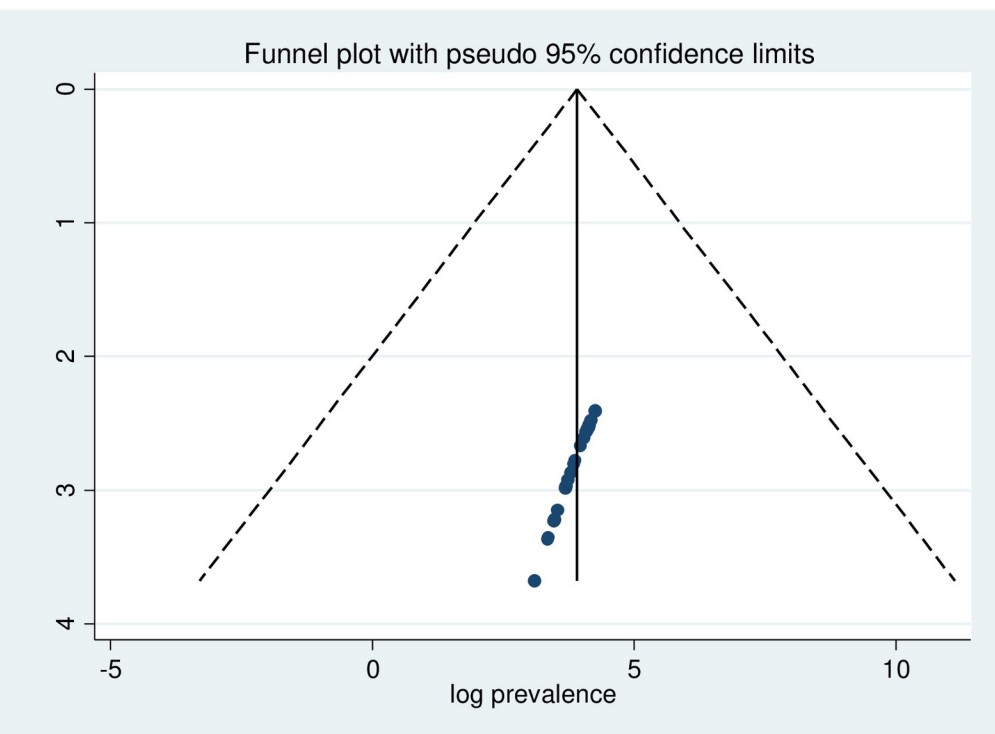

**Fig 3. Funnel plot of anxiety among medical and surgical patients in Ethiopia.**

anxiety among medical patients was having poor social support. Medical patients with poor social support were 2.22 at risk of developing anxiety compared to those with strong social support [OR = 2.22(1.62, 3.05)].

Patients who had perceived stigma were more than four times more likely to have anxiety than their counterparts [OR = 4.25(1.97, 9.18)]. Having a family history of mental illness was also associated with anxiety among medical patients [OR = 1.86(1.21, 2.86)] (Fig 7).

## Discussion

This systematic review and meta-analysis synthesized findings of 23 primary investigations carried out among medical and surgical patients in Ethiopia, to ascertain the pooled prevalence and associated factors of anxiety among 8490 study participants.

The pooled prevalence of anxiety among medical and surgical patients in Ethiopia was 48.82% with a 95% CI (42.66, 54.99). This result was consistent with a global meta-analysis among patients undergoing surgery (48%)[8], and studies conducted in Nigeria (51%) [57], Rwanda (52.1%) [58], Vietnam (43%) [59], Pakistan(45.95%) [60], and Turkey (52.2%) [61]. The pooled estimate of this review was higher compared to studies conducted in Kenya (37%) [62], China(15%) [63], India (31%) [64], and Brazil (35%) [65]. The possible explanation for

**Table 2. Egger's tests of anxiety among medical and surgical patients in Ethiopia.**

| Std_Eff | Coef. | Std. Err. | t | P>t | [95% Conf. | Interval] |
|---|---|---|---|---|---|---|
| Slope | 3.332214 | 19.97388 | 0.17 | 0.869 | -38.20575 | 44.87018 |
| Bias | 17.74579 | 8.047242 | 2.21 | 0.039 | 1.0106638 | 34.48095 |

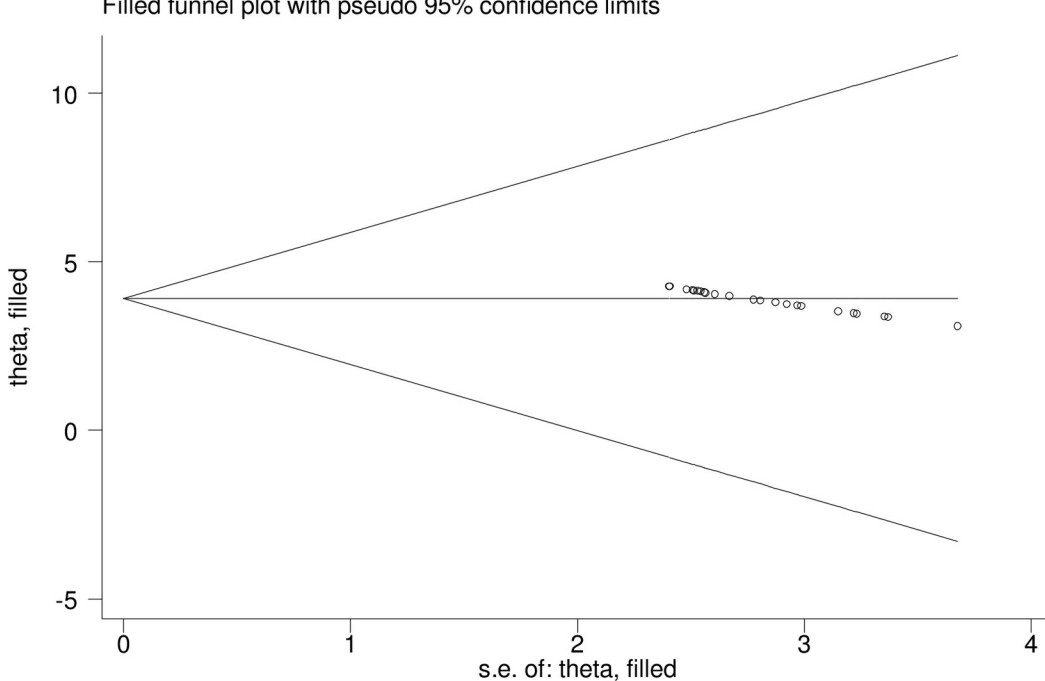

**Fig 4. Trim and fill analysis for meta-analysis on anxiety among medical and surgical patients in Ethiopia.**

this discrepancy could be due to sociocultural and socioeconomic differences. Moreover, access to information regarding their surgical procedure or medical illness, variations in the health care delivery system and tools used to screen anxiety might contributed to this inconsistency.

However, the pooled estimate of this study was lower than studies carried out in Tunisia 67.5% [66], Nepal (70.6%) [67], and the USA (72.7%) [22]. This variation could be due to the assessment tool differences and the type of case the study participants had. Studies conducted among patients undergoing surgical procedures that could have a greater impact on the increment of anxiety among this population may partly explain this variation. Furthermore, study

**Table 3. Subgroup analysis of anxiety and associated factors among medical and surgical patients in Ethiopia.**

| Variables | Subgroup | Number of studies | Sample size | Prevalence (95% CI) | I² (%) | P-value |
|---|---|---|---|---|---|---|
| **Region** | Amhara | 9 | 3173 | 48.70(38.34, 59.03) | 97.4 | ≤ 0.001 |
| | Addis Ababa | 6 | 2686 | 49.70(35.12, 64.29) | 98.2 | ≥ 0.001 |
| | Oromia | 4 | 1503 | 54.11(40.69, 54.99) | 96.6 | ≥ 0.001 |
| | *Others | 4 | 1542 | 42.59(28.83, 56.32) | 97 | ≥ 0.001 |
| **Case type** | Medical | 14 | 5685 | 43.63(35.79, 51.47) | 97.5 | ≥ 0.001 |
| | Surgical | 9 | 2800 | 56.97(49.79, 64.15) | 93.7 | ≥ 0.001 |
| **Assessment tool** | HADS | 11 | 4237 | 44.68(36.79, 52.57) | 96.6 | ≥ 0.001 |
| | S-STAI | 7 | 2126 | 57.55(51.30, 63.79) | 88.7 | ≥ 0.001 |
| | **Others | 5 | 2127 | 45.85(27.96, 63.74) | 98.8 | ≥ 0.001 |

*Others: SNNPs and Somalia.

**Others: PITI, GAD-7, BAIS, HAS, DAS.

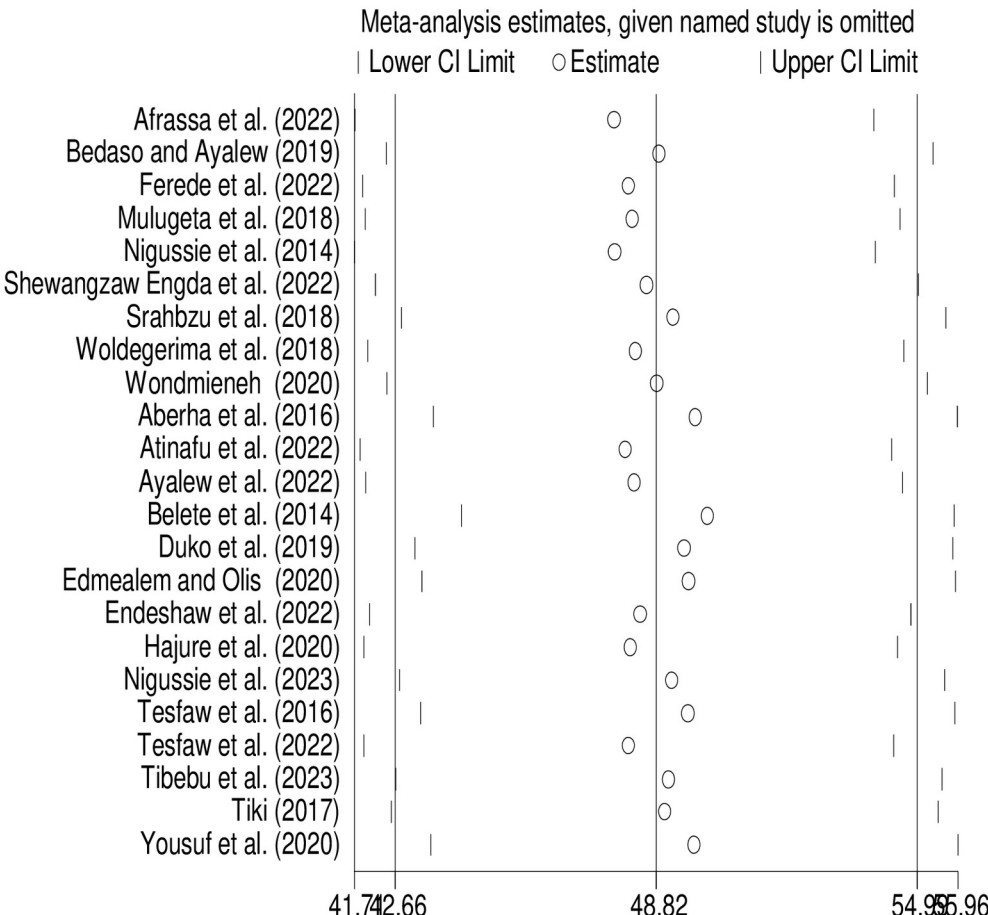

**Fig 5. A sensitivity analysis for studies included in this systematic review and meta-analysis.**

participants' variations for instance; the study conducted in the USA recruited only females who are more likely to experience anxiety compared to males.

In the subgroup analysis, the pooled prevalence of anxiety in Oromia region was higher which could be due to tool and case type differences. Furthermore, the prevalence of anxiety among studies assessed using S-STAI was higher compared to other tools. This could be a result of the psychometric properties of the tool and the tool was also used among patients undergoing surgery which could increase the pooled effect size of the anxiety.

When compared to medical patients, surgical patients were found to have a higher pooled prevalence of anxiety. This could be explained by the possibility that anxiety in those who are about to have surgery is more likely to be high due to emotional and physiological reactions to the anticipated surgery. Additionally, S-STAI was used in majority of studies including surgical cases, which might have indirectly influenced the results.

Regarding predictive variables to anxiety, in this systematic review and meta-analysis, females were more than two times more likely to develop anxiety than males among both medical and surgical patients. This finding was supported by studies conducted in Brazil [26], India [68], and China [69]. The possible reason for this might be women's affective responses to stressors, domestic violence, learned helplessness, hormonal changes associated with menstruation, lower income, and lower social status [70]. Besides this, the existence of specific

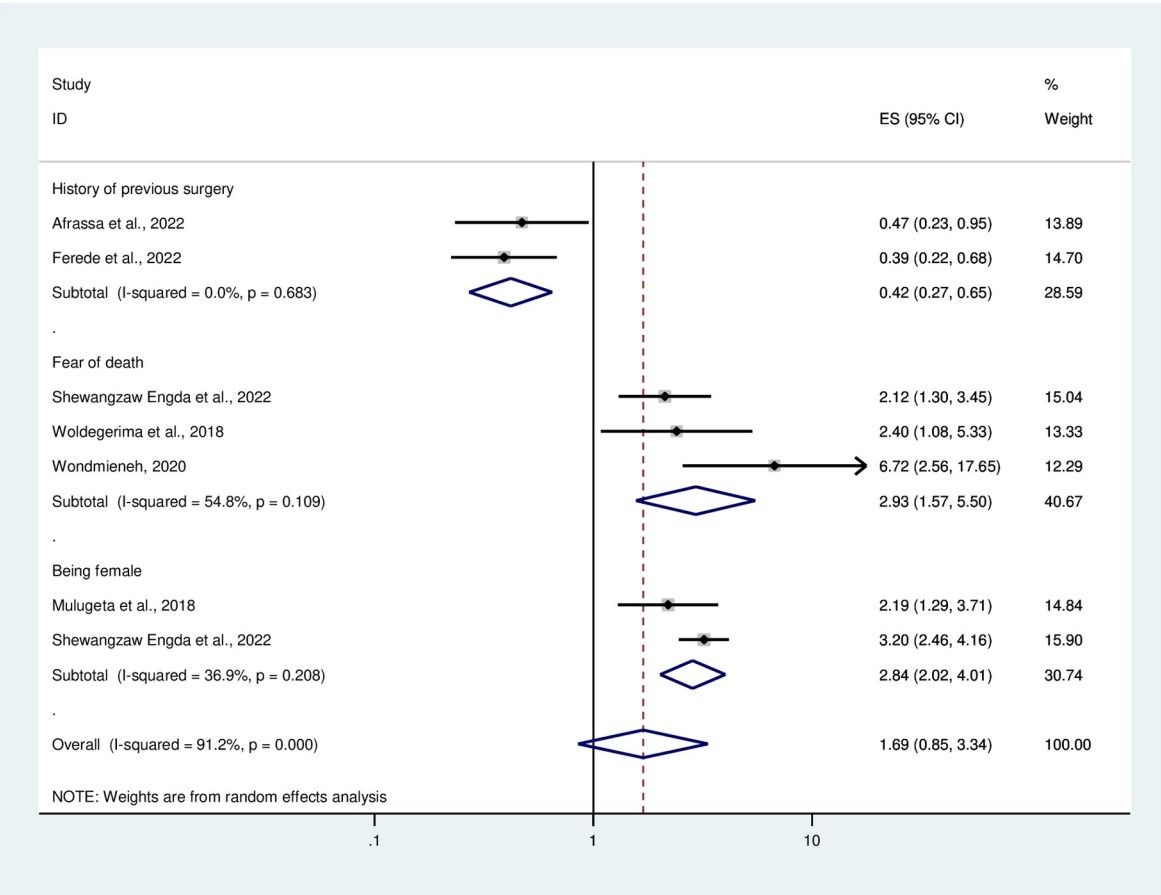

**Fig 6. The Forest plot showing factors associated with anxiety among surgical patients in Ethiopia.**

mental health conditions in women, such as premenstrual dysphoric disorder, postpartum depression, and postmenopausal mental health conditions, which are linked to hormonal changes, may raise the possibility for female patients to feel more anxious [71].

Surgical patients who had a fear of death to undergo surgery were around three times more likely to develop anxiety compared to their counterparts. This evidence was consistent with a survey done in the USA to assess fear of death and anesthesia complications in patients undergoing surgery [72]. This might be due to the anesthesia medication complications or the surgical procedure by itself patients perceive it would be unimaginable to wake up after the procedure is done [73].

This systematic review and meta-analysis also revealed that having poor social support was associated with anxiety among medical patients. This result was supported by studies in Nepal [74], USA [75], and Pakistan [76]. This could be possibly explained that the effect of getting insufficient emotional, instrumental and informational support from significant other which leads to cope poorly in stressful situations and decrease their resilience [77]. Medical patients who had perceived stigma were more than four times more likely to have anxiety than those who didn't have perceived stigma and this was consistent with studies in Malawi [78] and China [79]. This could be a result of stigmatized persons internalizing perceived prejudices and developing negative feelings about themselves and feeling ashamed and embarrassed

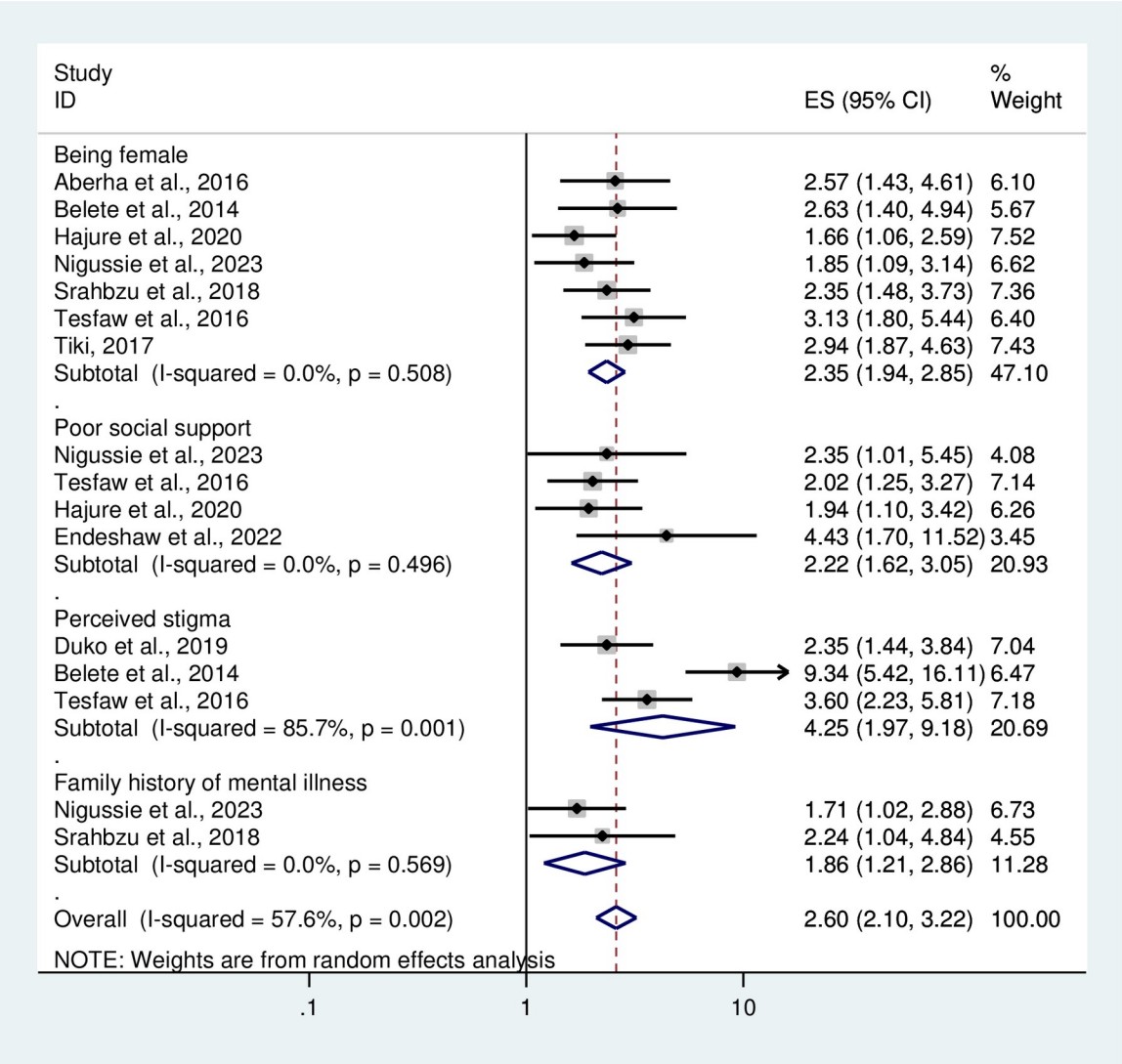

**Fig 7. The Forest plot showing factors associated with anxiety among medical patients in Ethiopia.**

about having the debilitating illness and bother about their social life which could prone them to develop anxiety [80].

Having a family history of mental illness was also an independent variable found to be associated with anxiety among medical patients. This finding was supported by a study conducted in Nepal [74]. A family history of mental illnesses is one risk factor for mental disorders (the hereditary effect), notably mood disorders, which carries a risk of between 10 and 25 percent for a child if one parent has a mood disorder [81].

## Limitations of the study

Although this systematic review and meta-analysis has a great advantage, in giving the pooled effect of anxiety and its determinants among medical and surgical patients in Ethiopia it has

limitations. The studies included in this review were all cross-sectional studies in which this finding couldn't show the temporal relationship of anxiety and its determinants.

## Conclusion and recommendation

The pooled prevalence of anxiety among medical and surgical patients in Ethiopia was found to be 48.82%. Fear of death and history of surgery among surgical patients, poor social support, having perceived stigma, and family history of mental illness among medical patients were associated with anxiety whereas being female was associated with outcome variable in both medical and surgical patients. Therefore, health professionals should routinely assess patients for anxiety besides treating medical and surgical illnesses to achieve the ultimate goal of patient care and to impact positively on the prognosis of these conditions.

## Supporting information

**S1 File. PRISMA-2020 checklist for anxiety.**
(XLSX)

**S2 File. Quality assessment scores for included studies.**
(DOCX)

**S3 File. Extracted data for anxiety.**
(DOCX)

## Acknowledgments

Our gratitude goes to the authors who conducted the primary investigations included in this systematic review and meta-analysis.

## Author Contributions

**Conceptualization:** Gidey Rtbey, Mamaru Melkam.

**Formal analysis:** Gidey Rtbey, Fantahun Andualem.

**Methodology:** Gidey Rtbey, Milen Mihertabe, Fantahun Andualem, Mamaru Melkam, Girmaw Medfu Takelle, Techilo Tinsae, Setegn Fentahun.

**Writing – original draft:** Gidey Rtbey.

**Writing – review & editing:** Gidey Rtbey, Milen Mihertabe, Fantahun Andualem, Mamaru Melkam, Girmaw Medfu Takelle, Techilo Tinsae, Setegn Fentahun.

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
