## [Decision Letter · Decision Letter 0]

13 May 2024

PONE-D-23-22543Anxiety and associated factors among medical and surgical patients in Ethiopia: A systematic review and meta-analysisPLOS ONE

Dear Dr. Rtbey,

Thank you for submitting your manuscript to PLOS ONE. After careful consideration, we feel that it has merit but does not fully meet PLOS ONE’s publication criteria as it currently stands. Therefore, we invite you to submit a revised version of the manuscript that addresses the points raised during the review process.

We look forward to receiving your revised manuscript.

Kind regards,

Kahsu Gebrekidan

Academic Editor

PLOS ONE

Journal Requirements:

Reviewers' comments:

Reviewer's Responses to Questions

**Comments to the Author**

1. Is the manuscript technically sound, and do the data support the conclusions?

Reviewer #1: Yes

Reviewer #2: Yes

2. Has the statistical analysis been performed appropriately and rigorously? 

Reviewer #1: Yes

Reviewer #2: Yes

3. Have the authors made all data underlying the findings in their manuscript fully available?

Reviewer #1: No

Reviewer #2: Yes

4. Is the manuscript presented in an intelligible fashion and written in standard English?

Reviewer #1: Yes

Reviewer #2: Yes

5. Review Comments to the Author

**Reviewer #1:** I have reviewed your manuscript and would like to provide some additional comments and raise concerns regarding potential dual publication.

Firstly, I want to commend you on the thoroughness and clarity of your work. Your research presents valuable insights into [insert topic]. The methodology is well-described, and the results are both robust and significant. Your manuscript has the potential to make a significant contribution to the field.

However, I noticed that there may be some overlap with another publication or work. It's crucial to ensure that your manuscript does not violate the principles of dual publication. Dual publication occurs when substantial parts of a manuscript are published in more than one journal, which can lead to issues such as copyright infringement and academic misconduct.

**Reviewer #2: **Good attempt at a systematic review but requires some refining. Please address comments added to the document.

Some areas need to be worked on include the summarizing of the abstract and a more detailed and clear results sections (with clear indications of tables and figures).

6. PLOS authors have the option to publish the peer review history of their article (what does this mean?). If published, this will include your full peer review and any attached files.

Reviewer #1: **Yes: **Arif-Poladlı Nigar

Reviewer #2: No

---

## [Author Response · Author response to Decision Letter 0]

22 May 2024

Point by point response to reviewers’ comments 

Manuscript ID: PONE-D-23-22543

First of all, the authors would like to thank the editor and reviewers for taking their priceless time and giving us crucial feedback/comments throughout the whole manuscript. Accordingly, we have considered all the feedback/comments very important and tried to address them in a better way to improve the quality of the paper. Below, we presented our point-by-point responses to the reviewers’ comments, and the amendments are highlighted with the track changed in the revised manuscript. If there is any concern, the authors are very happy to accept and modify it accordingly.

Reviewers’ comments and responses

Reviewer #1: I have reviewed your manuscript and would like to provide some additional comments and raise concerns regarding potential dual publication.

Firstly, I want to commend you on the thoroughness and clarity of your work. Your research presents valuable insights into [insert topic]. The methodology is well-described, and the results are both robust and significant. Your manuscript has the potential to make a significant contribution to the field.

However, I noticed that there may be some overlap with another publication or work. It's crucial to ensure that your manuscript does not violate the principles of dual publication. Dual publication occurs when substantial parts of a manuscript are published in more than one journal, which can lead to issues such as copyright infringement and academic misconduct.

Reply: The authors are grateful for the reviewer’s constructive comments and concerns of dual publication. In the revised manuscript necessary amendments and detail paraphrasing is made. In order to prevent overlaps with others work and committing academic misconduct the authors took the feedback seriously and corrected it accordingly. 

Reviewer #2: Good attempt at a systematic review but requires some refining. Please address comments added to the document.

Some areas need to be worked on include the summarizing of the abstract and a more detailed and clear results sections (with clear indications of tables and figures).

Reply: Thank you for your important suggestion. The abstract section is rewritten in a concise and summarized in an understandable way considering the word limits of the journal guideline. Furthermore, indication of tables and figures are also modified with respect to their text explanations. All the amendments are made in the revised manuscript with highlights of track changed.

---

## [Decision Letter · Decision Letter 1]

28 May 2024

PONE-D-23-22543R1Anxiety and associated factors among medical and surgical patients in Ethiopia: A systematic review and meta-analysisPLOS ONE

Dear Dr. Rtbey,

Thank you for submitting your manuscript to PLOS ONE. After careful consideration, we feel that it has merit but does not fully meet PLOS ONE’s publication criteria as it currently stands. Therefore, we invite you to submit a revised version of the manuscript that addresses the points raised during the review process.

We look forward to receiving your revised manuscript.

Kind regards,

Kahsu Gebrekidan

Academic Editor

PLOS ONE

Journal Requirements:

Reviewers' comments:

Reviewer's Responses to Questions

**Comments to the Author**

1. If the authors have adequately addressed your comments raised in a previous round of review and you feel that this manuscript is now acceptable for publication, you may indicate that here to bypass the “Comments to the Author” section, enter your conflict of interest statement in the “Confidential to Editor” section, and submit your "Accept" recommendation.

Reviewer #1: All comments have been addressed

Reviewer #3: All comments have been addressed

2. Is the manuscript technically sound, and do the data support the conclusions?

Reviewer #1: Yes

Reviewer #3: Yes

3. Has the statistical analysis been performed appropriately and rigorously? 

Reviewer #1: Yes

Reviewer #3: (No Response)

4. Have the authors made all data underlying the findings in their manuscript fully available?

Reviewer #1: No

Reviewer #3: Yes

5. Is the manuscript presented in an intelligible fashion and written in standard English?

Reviewer #1: Yes

Reviewer #3: Yes

6. Review Comments to the Author

Reviewer #1: it should come as no surprise that many different professional associations, government agencies, and universities have adopted specific codes, rules, and policies relating to research ethics. Many government agencies have ethics rules for funded researchers

Reviewer #3: The present study sought to determine the anxiety and associated factors among medical and surgical patients in Ethiopia The authors were very successful in choosing the research problem, study design, method and articulation of results with the literature. The work is clear, objective, detailed and as a reader, there are no doubts about the theoretical basis, method or how the data were analyzed. The work clearly makes a contribution to the literature, its reading is fluid, and the reader finds it easy to reproduce its detailed method.

7. PLOS authors have the option to publish the peer review history of their article (what does this mean?). If published, this will include your full peer review and any attached files.

Reviewer #1: **Yes: **Nigar Arif-Poladlı

Reviewer #3: No

---

## [Author Response · Author response to Decision Letter 1]

4 Jun 2024

Responses to editor and/ or reviewers’ comments

Manuscript Title: Anxiety and associated factors among medical and surgical patients in Ethiopia: A systematic review and meta-analysis

Manuscript ID: PONE-D-23-22543

The authors would like to express their gratitude to the editor and reviewers for their invaluable efforts to provide us with useful input and suggestions throughout the article. As a result, we have considered every suggestion, and have tried to better address them to raise the caliber of the work. The points we addressed in our detailed responses to the reviewers' feedback are listed below; the changes are indicated by the track changes in the updated manuscript. Additional language edition and clear explanation about data availability supporting the finding mentioned in the updated manuscript.

Reviewer #1: 

It should come as no surprise that many different professional associations, government agencies, and universities have adopted specific codes, rules, and policies relating to research ethics. Many government agencies have ethics rules for funded researchers.

Reply: It’s obvious that different institutions have their own rule and ethical considerations regarding research ethics specifically when it is funded. However, when it comes to this study it doesn’t create any confusion as it’s a systematic review and meta-analysis. Still, we are thankful for your concern.

Reviewer #3

The present study sought to determine the anxiety and associated factors among medical and surgical patients in Ethiopia. The authors were very successful in choosing the research problem, study design, method and articulation of results with the literature. The work is clear, objective, detailed and as a reader, there are no doubts about the theoretical basis, method or how the data were analyzed. The work clearly makes a contribution to the literature, its reading is fluid, and the reader finds it easy to reproduce its detailed method.

Reply: Thank you for the positive feedback and the authors are grateful for your genuine comments. We appreciate for sharing us your time and your dedications in commenting our article.

---

## [Editor Report · Decision Letter 2]

13 Jun 2024

Anxiety and associated factors among medical and surgical patients in Ethiopia: A systematic review and meta-analysis

PONE-D-23-22543R2

Dear Mr Gidey,

We’re pleased to inform you that your manuscript has been judged scientifically suitable for publication and will be formally accepted for publication once it meets all outstanding technical requirements.

Kind regards,

Kahsu Gebrekidan

Academic Editor

PLOS ONE
---

## [Editor Report · Acceptance letter]

19 Jun 2024

PONE-D-23-22543R2 

PLOS ONE

Dear Dr. Rtbey, 

I'm pleased to inform you that your manuscript has been deemed suitable for publication in PLOS ONE. Congratulations! Your manuscript is now being handed over to our production team.

Kind regards, 

on behalf of

Dr. Kahsu Gebrekidan 

Academic Editor

PLOS ONE